Early radiological adjacent segment changes following L4/5 fusion: a retrospective comparative study of oblique lateral interbody fusion and minimally invasive transforaminal lumbar interbody fusion

Liu Lantao 1
Ma Chicheng 2
Jiang Lianghai 1
Chen Longwei 1
Zhou Xinpeng 3
Wang Dechun spineking@163.com 1
1 Department of Spinal Surgery, Qingdao Municipal Hospital Group , Qingdao , Shandong Province , China
2 The Third Department of Orthopedics, The First Affiliated Hospital, Harbin Medical University , Harbin , Heilongjiang , China
3 Department of Spinal Surgery, Qingdao Municipal Hospital, Qingdao University , Qingdao , Shandong Province , China
Kass Philip
Electronic publication date: 2025 Aug 21
Publication date: 2025
Volume: 13
Electronic Location ID: e19918
Received 2025 Feb 13; Accepted 2025 Jul 23
Copyright: ©2025 Liu et al.
Copyright year: 2025
Copyright holder: Liu et al.
License: This is an open access article distributed under the terms of the Creative Commons Attribution License, which permits unrestricted use, distribution, reproduction and adaptation in any medium and for any purpose provided that it is properly attributed. For attribution, the original author(s), title, publication source (PeerJ) and either DOI or URL of the article must be cited.
License URL: https://creativecommons.org/licenses/by/4.0/

Keywords: Adjacent segment disease, Oblique lateral interbody fusion, Minimally invasive transforaminal lumbar interbody fusion, Radiological changes

Funding: The Medical and Health Technology Development Programs of Shandong Province No. 202304070277 This work was supported by the Medical and Health Technology Development Programs of Shandong Province (No. 202304070277). The funders had no role in study design, data collection and analysis, decision to publish, or preparation of the manuscript.

==============================
Background

Adjacent segment disease (ASD) has become one of the most common complications after lumbar interbody fusion. To date, there have been few reports on the radiological effects between oblique lateral interbody fusion (OLIF) and minimally invasive transforaminal lumbar interbody fusion (MIS-TLIF) in lumbar degenerative diseases adjacent to the superior and inferior segments.

Method

The data of patients treated with OLIF or MIS-TLIF due to L4/5 degenerative lumbar diseases from October 2018 to March 2022 were retrospectively analyzed. The anterior disc height (ADH), posterior disc height (PDH), intervertebral foramen height (FH), canal anteroposterior diameter (APD), foraminal area (FA), dural sac area (DSA), disc angle (DA), segmental lordosis angle (SLA), and lumbar lordosis (LL) were compared. The incidences of adjacent segment intervertebral disc degeneration (ADD), adjacent articular facet degeneration (AFD), and adjacent segment disease (ASD) were analyzed. Multiple linear regression analysis was performed on DSA of L3/4, L5/S1, and DA of L5/S1. Clinical assessments were performed using the visual analogue scale (VAS) and Oswestry disability index (ODI).

Results

At the final follow-up, the Δ SLA, Δ ADH, Δ PDH, Δ FH, Δ FA, and Δ LL of L4/5 in the OLIF group were significantly improved compared with those in the MIS-TLIF group (P < 0.05). The Δ DSA and Δ APD of L3/4 and L5/S1 segments and L5/S1 Δ DA in the OLIF group were improved significantly (P < 0.05). Meanwhile, multiple linear regression analysis showed that Δ L5/S1DA increased with the increase of Δ LL, and decreased with the increment of Δ L5/S1DA (Pre). The incidence of ADD and AFD in L3/4 of the OLIF group was higher than that of the MIS-TLIF group (P < 0.05). The incidence of ASD in the L5/S1 was lower than in the L3/4 level (P < 0.05).

Conclusion

Compared with MIS-TLIF, OLIF has advantages in improving lumbar sagittal balance, significantly decreasing degeneration of intervertebral discs and facet joints in adjacent segments in the early stage.

Introduction

Lumbar interbody fusion is a well-established surgical procedure for patients with degenerative disc disease who have failed conservative treatments and achieved satisfactory clinical results (Mobbs et al., 2015). From the traditional open approaches, such as posterior lumbar interbody fusion (PLIF) and transforaminal lumbar interbody fusion (TLIF), to the more current minimally invasive techniques, such as minimally invasive transforaminal lumbar interbody fusion (MIS-TLIF) and oblique lateral interbody fusion (OLIF), have achieved promising clinical outcomes. However, the biomechanics of the lumbar spine adjacent to the fusion level may be altered after interbody fusion. It has been shown that the adjacent lumbar disc bears greater loads, increases intervertebral disc pressure, and compensates for increased motion, resulting in further degeneration of the adjacent disc and facet joint and eventually leading to adjacent segment disease (ASD) (Sears et al., 2011; Masevnin et al., 2015). As a result, ASD has become one of the most common complications after lumbar interbody fusion.

It has been observed that the incidence of ASD after lumbar fusion might range from 4.7% to 27.4% in the literature (Mesregah et al., 2022). ASD was classified as adjacent segment disease (ASDis) and adjacent segment degeneration (ASDeg). In the definition of ASDis, new degenerative changes at a spinal level close to a level or levels of the spine that have undergone surgery are characterized by associated symptoms (radiculopathy, myelopathy, or instability). ASDeg is a term for radiographic alterations without accompanying symptoms (Riew et al., 2012). ASDis was derived from ASDeg (Lund & Oxland, 2011). Based on previous studies, strategies to prevent ASD development include adequate restoration of the spinal sagittal alignment, including segmental lordosis and lumbar lordosis, stabilization of the posterior ligament complex, and restriction of adjacent segmental injury (Takeda et al., 2022; Berven & Wadhwa, 2018; Park et al., 2004; Min et al., 2008). OLIF and MIS-TLIF are two of the mainstay minimal invasive modalities for the treatment of lumbar degenerative disc diseases and represent typical surgical approaches for indirect and direct decompression, respectively. OLIF was similar to or superior to MIS-TLIF in improving lumbar sagittal alignment (Yingsakmongkol et al., 2022; Gao et al., 2022). Recently, lateral lumbar interbody fusion (LLIF) and extreme lateral interbody fusion (XLIF) have become popular techniques for treating spinal illnesses such as degenerative lumbar disease and spondylolisthesis because of the indirect decompression and improved lumbar sagittal alignment. Compared with TLIF, LLIF had better clinical outcomes, such as less estimated blood loss, shorter operative time, a more significant improvement in lower back pain, and reserving posterior support elements (Hiyama et al., 2020). However, a systematic review demonstrated that MIS-TLIF had better ODI, VAS pain, and lower complication rates when compared to LLIF (Keorochana et al., 2018). There are controversies not only in clinical outcomes but also in the occurrence of ASD in the treatment of lumbar degenerative disease with LLIF and TLIF or PLIF. Min, Jang & Lee (2007) reported that the incidence of ASD was markedly reduced with the LLIF approach compared with conventional PLIF. However, Otsuki et al. (2023) believed that the incidence of ASD was not significantly reduced in the LLIF group compared to the TLIF group. Compared with LLIF and XLIF, the psoas major muscle is not cut off, and the lumbar plexus nerve is avoided from injury in the OLIF approach (Bamps et al., 2023; Hah & Kang, 2019). Thus, OLIF is widely used to treat lumbar degenerative disease.

There are few reports on the radiological changes between OLIF and MIS-TLIF in lumbar degenerative diseases adjacent to the superior and inferior segments. Therefore, we aimed to analyze the radiographic differences of the adjacent segments after OLIF in comparison to an age-, sex-, and segment-matched cohort of patients undergoing MIS-TLIF for the treatment of lumbar degenerative disease in the early stage. This study may provide a clinical observational basis for the development of ASD.

Materials & Methods

Ethical statement

This study was approved by the institutional review board of the ethics committee of Qingdao Municipal Hospital (No. 2023Y044) and was conducted in accordance with the Declaration of Helsinki. Initially, the participants were informed about the study’s purpose, scope, and data usage, which would be kept confidential and used only for scientific purposes. All patients would be informed that it was voluntary and that they could withdraw from the study at any time, and there were no payments or fees. At all stages of the study, patients were involved to get feedback about the comprehensibility of the patient information accompanying the informed consent. All participants provided written informed consent prior to participation.

Patient selection

The study is an age-, sex-, and segment-matched cohort study. We retrospectively analyzed the data of patients with lumbar degenerative disc disease treated with MIS-TLIF or OLIF at our hospital between October 2018 and March 2022. All the data collection and analysis began on 15 July 2023. Inclusion criteria were as follows: (1) failure to conservative treatment for more than 3 months; (2) lumbar instability or lumbar disc degeneration with a Pfirrmann grade of more than IV (Pfirrmann et al., 2001); (3) MIS-TLIF or OLIF with pedicle screw fixation at single L4/5; and (4) at least 10 months post-operative follow-up. Exclusion criteria were as follows: (1) previous history of lumbar surgery; (2) lumbar disease associated with malignancy, trauma, or infection; and (3) OLIF combined with posterior decompression. The two cohorts were matched by sex, age, and segment. According to the aforementioned inclusion and exclusion criteria, 25 patients from the OLIF group and 25 from the MIS-TLIF group were each included in the study (Fig. 1). The main surgical procedure was performed by a senior surgeon. One of the investigators will contact patients who meet the inclusion criteria to complete the related clinical questionnaires.

Figure 1 Flow chart for patient population, grouping, intervention and follow-up method.

Intervention

MIS-TLIF group

The patient is placed in a prone position under general anesthesia. The responsible segment (L4/5) was determined under C-arm. After routine skin disinfection, a quadrant expandable channel was placed on the side with severe symptoms and the position was confirmed under C-arm. The soft tissue on the surface of the articular process and lamina is removed. Under the microscope, part of the lamina, superior and inferior articular processes, and ligamentum flavum were excised. Expose the intervertebral disc and nerve root. An appropriate cage was placed after the intervertebral disc was removed. Then, the pedicle screws and rods were placed on the decompression side. The operating table is tilted at 45 degrees, and the microscope is adjusted. The base of the spinous process and the contralateral inner layer of the lamina were blurred away. The contralateral nerve root was exposed after the ligamentum flavum was removed. The pedicle screws and rods were fixed under the channel on the opposite side. Wound drainage was applied.

OLIF group

OLIF was performed as reported previously (Hu et al., 2021). In brief, the right-side lateral decubitus posture was used after general anesthesia. The spine was checked for rotation using anterior-posterior fluoroscopy, and the target intervertebral disc space’s central point was checked using lateral fluoroscopy. A 4-cm long incision was made 3.0–5.0 cm in front of the central point of the target intervertebral disc. Three muscular layers were bluntly split along the direction of muscle fibers until the retroperitoneal fat was exposed. The fat tissues were peeled off up to the front edge of the psoas muscle using peanuts. A working retractor was fixed to the upper vertebral body with a pin. A discectomy was performed and a peek cage (six degrees, 18 mm width) filled with artificial bone was inserted into the intervertebral space. After that, the patient was transferred to a prone position to undergo posterior percutaneous pedicle screw instrumentation. The drainage was used if necessary.

Data definitions and collection

Radiographic evaluation

Preoperative and final follow-up radiographs and magnetic resonance imaging (MRI) scans were performed at the supine position and evaluated using the PACS software. Anterior disc height (ADH), posterior disc height (PDH), foramen height (FH), foramen area (FA), segmental lumbar angle (SLA), disc angle (DA), and lumbar lordosis (LL) were measured on the lateral X radiographs (Jin et al., 2019) (Fig. 2). The ADH was radiographically defined as the vertical distance between the anterior margins of the superior vertebra’s inferior endplate and the inferior vertebra’s superior endplate, measured on standardized lateral radiograph. The PDH was measured as the vertical distance between the posterior margins of the superior vertebra’s inferior endplate and the inferior vertebra’s superior endplate on standardized lateral radiograph. FH was measured on a lateral radiograph as the longest distance between the lower margin of the upper pedicle and the upper margin of the lower pedicle. FA was defined as the area bounded by the facet joint surface anteriorly, the posterosuperior portion of the inferior vertebral body, the surface of the intervertebral disk posteriorly, the superior and inferior adjacent vertebral pedicles, the posteroinferior portion of the superior vertebral body, and the superior and inferior adjacent vertebral pedicles. SLA was calculated as the angle between tangent lines to the inferior endplate of the vertebra below the fused segment and the superior endplate of the vertebra above the fused segment. DA was defined as the angle between tangent lines to the superior and inferior endplates of the same disc. LL was defined as the angle between tangent lines to the superior of L1 and S1 superior endplates. Dural sac area (DSA), anteroposterior canal diameter (APD), the Pfirrmann classification of the intervertebral disc, and the Weishaupt classification of the articular process were measured on the MRI (Jin et al., 2019) (Fig. 3). DSA was defined as the area of the transverse dura capsule at the mid-level of the disc measured by axial T2-weighted MRI. APD was defined as the canal’s width in the disc’s median section measured by sagittal T2-weighted MRI. Adjacent disc degeneration (ADD) was defined as a Pfirrmann grade of ≥1 degenerative change from preoperative to final follow-up (Stosch-Wiechert et al., 2022). The adjacent facet joint degeneration (AFD) was defined as a degeneration grade of ≥1 from preoperative to final follow-up according to the Weishaupt classification (Weishaupt et al., 1999). All the preoperative and post-operative imaging data were evaluated by two independent spine surgeons. If the bias was significant, the decision was made by a third senior physician.

Figure 2 Overview of the radiographic parameters of interest.

Line segments a and b indicated PDH and ADH; Green arrow indicated FH; Yellow area indicated FA; The angle between red tangent lines of c and d indicated SLA; The angle between blue tangent lines of e and f indicated LL; The angle between yellow arrows indicated DA.

Figure 3 Measurement of the radiographic parameters of ASC on MRI.

Red line segment indicated APD; Yellow area indicated DSA.

Clinical evaluation

Clinical outcomes were assessed using the visual analog scale (VAS), and Oswestry disability index (ODI) preoperatively and at the final follow-up (Rauh, Andersen & Rosenberg, 2013; Martin et al., 2019).

Statistical processing and analysis

All statistical analyses were performed using SPSS Statistics Version 25.0 (IBM, Armonk, NY, USA). Data were screened for abnormalities and normalities using the Shapiro–Wilk test and a dichotomous plot. The Student-t test was used for the analysis of groups with normally distributed continuous variables. The Mann–Whitney U test was used for the intergroup analysis of discrete variables, categorical variables, and continuous variables that were not normally distributed. The categorical variables were tested using Fisher’s exact test with chi-squares. Multiple linear regression was used to examine the relationship between ΔL5/S1DA and ΔL4/5ADH, ΔL4/5PDH, ΔSLA, ΔLL, ΔL5/S1DA (Pre) and between ΔL5/S1 DSA, ΔL3/4 DSA and ΔL4/5ADH, ΔL4/5PDH, ΔSLA, ΔLL. P < 0.05 was considered statistically significant.

Results

Demographics of patients

According to the inclusion and exclusion criteria, 63 patients met the criteria, and five patients declined to participate in the study. Then 30 patients enrolled in the MIS-TLIF group and 28 patients enrolled in the OLIF group. Finally, five patients in the MIS-TLIF group and three patients in the OLIF group were lost in the follow-up (Fig. 1). A total of 25 patients, seven males and 18 females, were treated with OLIF, with a mean follow-up of 13.12 ± 3.24 months. 25 patients, 10 males and 15 females, were treated with MIS-TLIF, with a mean follow-up of 17.28 ± 8.68 months. The mean age and gender distributions were similar between the MIS-TLIF and OLIF groups. There were no significant differences in other baseline demographic characteristics, including the BMI, follow-ups, and diagnosis (Table 1). There were no differences in the L3/4 and L5/S1 of the ADD and AFD between the OLIF and MIS-TLIF (Table 1).

Table 1 Summary of demographic features.

	OLIF group (n = 25)	MIS-TLIF group (n = 25)	P value	
Age (Year)	63.12 ± 7.84	64.12 ± 8 .79	0.526	
Sex (Male/Female)	7/18	10/15	0.551	
BMI	25.59 ± 2.93	26.23 ± 2.42	0.393	
Preoperative diagnosis		0.569	
Lumbar disc herniation	2	4		
Degenerative spondylolisthesis	4	2		
Lumbar spinal stenosis	19	19		
Follow-up time (Month)	13.12 ± 3.24	17.28 ± 8.68	0.131	
Pfirrmann grade (pre-operation)				
L3/4			0.456	
II	13	15		
III	12	9		
IV	0	1		
L5/S1			0.355	
II	12	10		
III	8	13		
IV	5	2		
Weishaupt classification (pre-operation)				
L3/4			0.469	
0	15	12		
1	10	12		
2	0	1		
L5/S1			0.470	
0	9	11		
1	16	13		
2	0	1		

Radiographic outcomes

As shown in Table 2, there was no significant difference in preoperative ADH, PDH, FH, FA, SLA, or LL in the L4/5 surgical segment. The OLIF group had a substantially higher mean increase in ΔADH and ΔPDH at the final follow-up visit (4.46 ± 3.03 mm and 2.29 ± 1.34 mm) than the MIS-TLIF group (1.19 ± 1.86 mm and 0.77 ± 1.04 mm, P < 0.01). In the OLIF group, FH was higher than in the MIS-TLIF group (21.9 ± 83.48 mm vs. 20.33 ± 2.43 mm, P = 0.046), and the mean increase of ΔFH was higher in the OLIF group (3.31 ± 2.97 mm vs 1.82 ± 1.85 mm, P = 0.010). The OLIF group’s mean ΔFA increase was higher than the MIS-TLIF group’s (48.42 ± 27.49 mm2 vs 32.10 ± 27.64 mm2, P = 0.029). The mean ΔSLA increase in the OLIF group was higher compared with the MIS-TLIF group’s (4.14 ± 3.23 degrees vs 1.83 ± 3.00 degrees, P = 0.017). Additionally, the mean increase in ΔLL was more significant in the OLIF group than in the MIS-TLIF group (6.90 ± 6.67 degrees vs 2.50 ± 4.58 degrees, P = 0.037).

Table 2 L4/5 radiological outcomes of OLIF group and MIS-TLIF group.

L4/5	OLIF group (n = 25)	MIS-TLIF group (n = 25)	P value	
Anterior disc height (mm)				
Preoperative	10.28 ± 3.73	12.47 ± 2.33	0.164	
At the last Follow-up	14.74 ± 2.70	13.66 ± 1.97	0.309	
Correction	4.46 ± 3.03	1.19 ± 1.86	<0.01**	
Posterior disc height (mm)				
Preoperative	5.36 ± 1.56	6.34 ± 1.06	0.145	
At the last Follow-up	7.65 ± 1.73	7.12 ± 1.18	0.079	
Correction	2.29 ± 1.34	0.77 ± 1.04	<0.01**	
Segment lordosis angle (°)				
Preoperative	13.64 ± 6.66	15.76 ± 6.23	0.963	
At the last Follow-up	17.77 ± 5.52	17.59 ± 5.95	0.770	
Correction	4.14 ± 3.23	1.83 ± 3.00	0.017*	
Foraminal height (mm)				
Preoperative	18.66 ± 3.63	18.51 ± 2.58	0.252	
At the last Follow-up	21.98 ± 3.48	20.33 ± 2.43	0.046*	
Correction	3.31 ± 2.97	1.82 ± 1.85	0.010*	
Foraminal area (mm2)				
Preoperative	144.11 ± 33.67	149.98 ± 36.53	0.823	
At the last Follow-up	192.53 ± 39.36	182.08 ± 43.74	0.248	
Correction	48.42 ± 27.49	32.10 ± 27.64	0.029*	
Cross-sectional area (mm2)				
Preoperative	87.31 ± 28.43	75.11 ± 17.89	0.097	
At the last Follow-up	153.35 ± 52.68	138.25 ± 39.52	0.327	
Correction	66.04 ± 41.18	63.14 ± 35.25	0.823	
Anteroposterior diameter (mm)				
Preoperative	7.99 ± 2.23	7.03 ± 1.11	0.133	
At the last Follow-up	11.85 ± 2.87	10.61 ± 2.40	0.146	
Correction	3.87 ± 2.28	3.58 ± 2.34	0.479	
Lumbar lordosis				
Preoperative	37.48 ± 15.03	39.29 ± 11.60	0.793	
At the last Follow-up	44.37 ± 10.67	41.78 ± 9.73	0.931	
Correction	6.90 ± 6.67	2.50 ± 4.58	0.037*	
Notes.

* indicated P < 0.05.

** indicated P < 0.01.

At the upper adjacent segment (L3/4), both the average increase in ΔDSA and ΔAPD were higher in the OLIF group than in the MIS-TLIF group (13.63 ± 9.90 mm2 vs 6.65 ± 9.03 mm2, P = 0.035 and 0.81 ± 1.05 mm vs 0.29 ± 0.90 mm, P = 0.046, respectively) at the final follow-up. There was no significant difference in ΔADH, ΔPDH, ΔDA, ΔFH, and ΔFA between the OLIF group and MIS-TLIF group (Table 3). At the lower adjacent segment (L5/S1), both the average increase in ΔDSA and ΔAPD were higher in the OLIF group than in the MIS-TLIF group (23.92 ± 12.76 mm2 vs 11.97 ± 17.08 mm2, P = 0.018 and 0.86 ± 0.88 mm vs 0.66 ± 0.93 mm, P = 0.030, respectively). Additionally, the OLIF group’s mean ΔDA increase was higher than the MIS-TLIF group’s (2.40 ± 2.59 degrees vs 0.90 ± 2.81 degrees, P = 0.031). No significant differences were observed in ΔADH, ΔPDH, ΔFH, and ΔFA between the OLIF group and the MIS-TLIF group (Table 4).

Table 3 L3/4 radiological outcomes of OLIF group and MIS-TLIF group.

L3/4	OLIF group (n = 25)	MIS-TLIF group (n = 25)	P value	
Anterior disc height (mm)				
Preoperative	11.64 ± 3.03	12.54 ± 2.86	0.431	
At the last Follow-up	12.69 ± 2.03	12.57 ± 2.29	0.605	
Correction	1.05 ± 1.89	0.03 ± 1.14	0.053	
Posterior disc height (mm)				
Preoperative	5.88 ± 1.52	6.81 ± 1.56	0.697	
At the last Follow-up	6.85 ± 1.64	6.71 ± 1.60	0.675	
Correction	0.97 ± 1.51	-0.10 ± 1.12	0.129	
Disc angle (°)				
Preoperative	8.17 ± 3.56	7.47 ± 2.94	0.255	
At the last Follow-up	9.64 ± 2.34	8.55 ± 2.50	0.987	
Correction	1.47 ± 1.95	1.08 ± 1.74	0.898	
Foraminal height (mm)				
Preoperative	22.60 ± 3.68	20.50 ± 2.42	0.113	
At the last Follow-up	22.85 ± 3.22	21.43 ± 2.92	0.074	
Correction	0.25 ± 3.00	0.93 ± 2.12	0.135	
Foraminal area (mm2)				
Preoperative	213.56 ± 55.31	192.10 ± 43.52	0.108	
At the last Follow-up	209.64 ± 48.34	188.92 ± 46.51	0.545	
Correction	−2.31 ± 23.87	−3.17 ± 13.68	0.125	
Cross-sectional area (mm2)				
Preoperative	120.30 ± 29.35	102.73 ± 25.22	0.546	
At the last Follow-up	133.93 ± 33.05	109.38 ± 30.32	0.500	
Correction	13.63 ± 9.90	6.65 ± 9.03	0.035*	
Anteroposterior diameter (mm)				
Preoperative	10.05 ± 1.67	8.58 ± 1.76	0.749	
At the last Follow-up	10.86 ± 1.69	8.87 ± 1.79	0.902	
Correction	0.81 ± 1.05	0.29 ± 0.90	0.046*	
Notes.

* indicated P < 0.05.

Table 4 L5/S1 radiological outcomes of OLIF group and MIS-TLIF group.

L5/S1	OLIF group (n = 25)	MIS-TLIF group (n = 25)	P value	
Anterior disc height (mm)				
Preoperative	12.85 ± 3.20	12.72 ± 2.54	0.206	
At the last Follow-up	13.89 ± 3.17	13.27 ± 2.69	0.210	
Correction	1.03 ± 1.70	0.55 ± 1.25	0.305	
Posterior disc height (mm)				
Preoperative	5.49 ± 1.43	5.84 ± 1.88	0.416	
At the last Follow-up	5.96 ± 1.61	6.33 ± 1.88	0.276	
Correction	0.48 ± 1.29	0.26 ± 1.02	0.355	
Disc angle (°)				
Preoperative	12.42 ± 3.51	12.68 ± 3.75	0.849	
At the last Follow-up	14.82 ± 1.94	13.58 ± 2.69	0.093	
Correction	2.40 ± 2.59	0.90 ± 2.81	0.031*	
Foraminal height (mm)				
Preoperative	16.10 ± 3.17	15.37 ± 2.11	0.337	
At the last Follow-up	15.87 ± 1.83	15.51 ± 1.78	0.852	
Correction	0.17 ± 1.73	0.13 ± 1.41	0.124	
Foraminal area (mm2)				
Preoperative	143.22 ± 27.57	132.55 ± 34.48	0.349	
At the last Follow-up	143.98 ± 25.99	134.08 ± 30.25	0.775	
Correction	0.75 ± 18.51	1.52 ± 15.61	0.961	
Cross-sectional area (mm2)				
Preoperative	149.27 ± 35.29	121.86 ± 38.18	0.850	
At the last Follow-up	173.19 ± 32.07	133.83 ± 39.20	0.303	
Correction	23.92 ± 12.76	11.97 ± 17.08	0.018*	
Anteroposterior diameter (mm)				
Preoperative	12.30 ± 1.82	10.59 ± 1.91	0.930	
At the last Follow-up	13.16 ± 1.95	11.25 ± 1.82	0.569	
Correction	0.86 ± 0.88	0.66 ± 0.93	0.030*	
Notes.

* indicated P < 0.05.

The incidence rates of ADD in L3/4, L5/S1, and overall in OLIF group were 4.0%, 4.0%, and 8.0%, respectively. The incidence rates of ADD in L3/4, L5/S1 and overall in the MIS-TLIF group were 28.0%, 12.0%, and 40.0%, respectively. The incidences of AFD in L3/4, L5/S1, and overall in the OLIF group were 4.0%, 8.0%, and 12.0%, respectively. The incidences of AFD in L3/4, L5/S1, and overall in the MIS-TLIF group were 24.0%, 16.0%, and 40.0%, respectively. The incidence of ADD and AFD at L3/4 in the OLIF group was significantly lower than that in the MIS-TLIF group (Table 5). The incidences of ASD in the OLIF group and MIS-TLIF group were 4.0% (1/25) and 12.0% (3/25), respectively (P = 0.297). There was no significant difference in the incidence of ASD between OLIF and MIS-TLIF (Table 5). The overall incidence of L3/4 ASD (8.0%, 4/50) was significantly higher than that of L5/S1 (0/50, P = 0.041). This result indicates that ASD is more likely to occur in the upper intervertebral disc.

Table 5 Comparison of adjacent segment degeneration between OLIF and MIS-TLIF group.

	OLIF group (n = 25)	MIS-TLIF group (n = 25)	P value	
Adjacent disc degeneration				
L3-4	1/25 (4%)	7/25 (28%)	0.021	
L5-S1	1/25 (4%)	3/25 (12%)	0.297	
Total	2/25 (8%)	10/25 (40%)	0.008	
Adjacent articular facet degeneration				
L3-4	1/25 (4%)	6/25 (24%)	0.042	
L5-S1	2/25 (8%)	4/25 (16%)	0.384	
Total	3/25 (12%)	10/25 (40%)	0.024	
Adjacent segment disease	1/25 (4%)	3/25 (12%)	0.297	

Multiple linear regression analysis showed that ΔL5/S1 DA increased with the increment of ΔLL and decreased with the increment of ΔL5/S1DA (Pre), and the regression model was ΔL5/S1DA = 0.151 × ΔLL − 0.541 × ΔL5/S1DA (Pre)+7.119 (P < 0.05, Table 4). Both ΔL3/4 DSA and ΔL5/S1 DSA increased with the increase of ΔSLA and ΔL4/5ADH, the regression models were ΔL3/4DSA = 1.411 × ΔSLA + 2.294 × ΔL4/5ADH+2.971 (P < 0.05, Table 3), and ΔL5/S1DSA = 3.291 × L4/5ΔSLA + 1.779 × Δ L4/5ADH+4.053 (P < 0.05, Table 4), respectively.

Clinical outcomes

Preoperative and final follow-up clinical outcomes are shown in Table 6. After OLIF or MIS-TLIF, VAS, and ODI scores significantly improved compared with preoperatively (P < 0.05). Preoperative and post-operative VAS and ODI scores did not differ substantially between the two groups (P > 0.05). During follow-up, one patient in the OLIF group experienced L3/4 ASD(is) and was treated conservatively (typical case in Fig. 4). Three patients in the MIS-TLIF group developed L3/4 ASD(is) during follow-up, one patient was treated with L3/4 stand-alone OLIF and two patients with L3/4 nerve block and other treatments conservatively (typical case in Fig. 5).

Table 6 Clinical outcomes of OLIF group and MIS-TLIF group.

	OLIF group (n = 25)	MIS-TLIF group (n = 25)	P value	
VAS-Lower back				
Preoperative	5.16 ± 0.94	5.16 ± 1.07	0.759	
At the last Follow-up	1.44 ± 0.51	1.76 ± 0.66	0.085	
VAS-Leg				
Preoperative	6.44 ± 1.08	6.24 ± 1.01	0.545	
At the last Follow-up	1.56 ± 0.71	1.68 ± 0.75	0.699	
ODI				
Preoperative	24.56 ± 7.26	25.92 ± 5.52	0.240	
At the last Follow-up	12.76 ± 4.92	12.76 ± 2.93	0.591	

Figure 4 Female, 71 years old, underwent L4/5 OLIF due to L4/5 spinal stenosis.

Twelve months after the operation, due to numbness of the lower limbs, she was diagnosed with L3/4 spinal stenosis and treated conservatively. (A) Preoperative lateral X-ray showed narrowing of L4/5 intervertebral space; (B and C) Sagittal and axial MRI showed L4/5 spinal canal stenosis; (D) showed preoperative L3/4 spinal canal was normal; (E) Lateral X-ray showed that the L4/5 intervertebral space was enlarged after operation; (F and G) Sagittal and axial MRI showed that the area of the L4/5 spinal canal was significantly enlarged after OLIF; (H) MRI showed that the L3/4 spinal stenosis after 12 months.

Figure 5 Female, 71 years old, underwent L4/5 MIS-TLIF due to lumbar spinal stenosis combined with lumbar disc herniation (A–D).

Nineteen months after L4/5 MIS-TLIF, due to low back pain and lower extremity pain, she was diagnosed with L3/4 instability and spinal stenosis (E–J). Standalone OLIF (L3/4) was performed again (K–L). A, Lateral X-ray showed lumbar degeneration; (B and C) Sagittal and axial MRI showed L4/5 spinal canal stenosis with intervertebral disc herniation; D, Axial MRI showed L3/4 spinal canal was normal; (E) Lateral X-ray showed widening of intervertebral space after MIS-TLIF; (F and G) Sagittal and axial MRI showed no obvious stenosis of the L4/5 spinal canal after postoperative 19 months; (H) Axial MRI showed mild stenosis of the L3/4 spinal canal after postoperative 19 months of L4/5 MIS-TLIF; (I and J) Lumbar flexion and extension X-ray indicated L3/4 instability; (K) Lateral X-ray showed cage position after L3/4 standalone OLIF; (L) Axial MRI showed the status of the L3/4 spinal canal 3 days after L3/4 standalone OLIF.

Discussion

ASD is one of the most frequent complications following lumbar interbody fusion and is also the primary reason for post-operative revision, accounting for 8.7% of reoperations (Irmola et al., 2018). Thus, further understanding of the changes associated with adjacent segments after lumbar interbody fusion is essential to avoid reoperations. Both OLIF and MIS-TLIF have been used extensively in treating lumbar degenerative disease with good clinical results (Garg & Mehta, 2019; Liu & Feng, 2020). However, little is known about the short-term morphological changes of OLIF and MIS-TLIF on adjacent segments assessed by radiographic imaging, so we conducted this study and obtained preliminary results. This study indicates that OLIF has advantages in improving lumbar sagittal balance, radiological parameters of adjacent segments, and decreasing degeneration of intervertebral discs and facet joints in adjacent segments compared with MIS-TLIF in the early stage. ASD is more likely to occur in the adjacent upper intervertebral disc.

In contrast to OLIF, which delivers indirect decompression to the surgical site while protecting the paravertebral muscles and posterior bony structures, MIS-TLIF delivers direct decompression through the intervertebral foramina into the spinal canal. OLIF avoids interfering with the spinal canal and nerve roots. Therefore, OLIF has a shorter hospital stay and less estimated blood loss (Lin et al., 2018), which aligns more with the conception of enhanced recovery after surgery and minimally invasive treatment. Similar to previous studies (Sheng et al., 2020), both OLIF and MIS-TLIF significantly reduced pain and improved limb function. Previous studies have shown that the incidence and revision rate of ASD in L4/5 segmental lumbar fusion increase steadily over time. The incidence at 3 years, 5 years, and 10 years after the initial operation are approximately 1%, 10%, and 25%, respectively (Heo et al., 2015). However, in our study, we observed a higher incidence of ASD (8%, 4/50), which could be attributed to our small sample size and the older age of our patients. First, as previous studies have shown, the age older than 60 years is an independent risk factor for ASD after lumbar fusion (Lee et al., 2014). In this study, the average age of both groups was over 60 years old, which undoubtedly increased the incidence of ASD. Second, patients in both groups had higher average BMIs, further sped up adjacent segment degeneration. Lastly, our sample size was relatively small, which also adds some chance to the higher incidence of ASD.

The results of the present study showed that the OLIF group was superior to the MIS-TLIF group in the improvement of ADH, PDH, FH, FA, SLA, and LL at the operative level, consistent with previous studies. According to Zhu et al. (2022) OLIF was superior to MIS-TLIF in restoring DH and LL. Meanwhile, Li et al. (2023) suggested that the combination of OLIF and posterior pedicle screw fixation achieved better lumbar lordosis correction than MIS-TLIF. Lin et al. (2018) and Yingsakmongkol et al. (2022) believed that compared with MIS-TLIF, OLIF has more advantages in improving FH and FA. SLA and LL recovery are essential in avoiding adjacent segment degeneration. Okuda et al. (2021) proposed that a decreased ΔSLA at the fused segment would cause an increase in range of motion (ROM) and load compensation at the adjacent fused segment, hence accelerating the degeneration at the adjacent segment. Matsumoto et al. (2017) concluded that achieving adequate LL was necessary to improve the spinal pelvis sagittal balance and prevent degeneration of adjacent segments after L4/5 single-segment surgery. Andrada discovered that the L4/5 kyphosis model induced higher mobility and disc strain (stress and morphology) at the adjacent level in a cadaveric model following single-level lumbar fusion (Pereira et al., 2022). In an in vitro biomechanical model, Lu & Lu (2019) demonstrated that OLIF increased the stress distribution in the endplate and cancellous bone at L4/5 compared to TLIF, which was advantageous for preventing the subsidence of the Cage and maintaining ADH, PDH, and SLA. Similar findings were found in our study, which found lower rates of ADD and AFD than in the MIS-TLIF group, and significantly higher increments in L4/5 SLA and LL were observed in the OLIF group compared to the MIS-TLIF group. The progression of ASD after indirect decompression with LLIF was also related to preexisting AFD reported by Ouchida et al. (2022). Placing a cage of a specific angle and size into the intervertebral space will lead to significant interspinal detachment and recovery of the segmental angles, improving the sagittal balance of the lumbar spine and allowing for full LL recovery. We hypothesized that the radiological progression of ASD would not be accelerated by the OLIF technique, as supported by reduced degenerative changes on postoperative imaging.

There are few studies on the effects of OLIF and MIS-TLIF on adjacent segment imaging and spinal alignment. According to Li, Tong & Wang (2022) the L5/S1 DA was significantly higher in the OLIF group compared to the TLIF group, and this increase was linearly associated with the rise in L4/5 IDH (intervertebral disc height) and fall in PT (pelvic tilt). Verst et al. (2023) studied the adjacent segment angle changes after lateral lumbar interbody fusion treatment for single-segment degenerative diseases. They found that the angle of the lower adjacent segment after single-segment surgery is significantly reduced compared to that before surgery, and there is a negative correlation between the angle of the lower adjacent segment and the angle of the preoperative lower adjacent segment and the correction angle of surgical segment angles (Verst et al., 2023). The present findings revealed that the increase in ΔDA in the inferior adjacent segment was tremendous in the OLIF group compared to the MIS-TLIF group. Additionally, multiple linear regression analysis demonstrated a positive correlation between this increase and improvement in lumbar lordosis while showing a negative correlation with the preoperative angle of the inferior adjacent segment. Some studies have suggested that improving the lordosis of the surgical segment will reduce the lordosis of the adjacent segment that was compensatory before surgery (O’Connor et al., 2022). This would attenuate the lordotic gain of the surgical segment, resulting in a mismatch between SLA and LL recovery. However, our results showed that the improvement in ΔLL was significantly more significant than ΔSLA in the OLIF group, which also indicated that the improvement in lordosis of the surgical segment caused changes in adjacent segments, which eventually led to the improvement of overall lumbar lordosis. The discrepancy may be attributed to the older age and inadequate compensatory capacity of adjacent segments in the patients included in our study. Additionally, based on radiographic indicators, the patient’s preoperative lower adjacent segment angle was smaller, potentially resulting in more remarkable post-operative recovery. Additionally, the entire lumbar spine is globally recalibrated because OLIF has a better recovery ability than MIS-TLIF due to the satisfied SLA and LL. This could help explain why OLIF restores lumbar overall sagittal balance better than MIS-TLIF in older adults with stiff degenerative lumbar disease. According to a recent study by Hiyama et al. (2023) Roussouly types I, II, and IV experienced significant improvements in their vertical axis and lumbar lordosis following LLIF surgery; however, the axial central canal area and midsagittal canal diameter of the thecal sac were smaller than those of Roussouly type III. Our results are not entirely consistent with their study. The following are some potential explanations for this discrepancy: ① Different inclusion criteria. The surgical segments included in our study are mainly L4/5 single segments, excluding L1/2, L2/3, L3/4, or multiple levels. Different fusion segments affect the angle of lumbar lordosis change (Kamalanathan et al., 2020). The position and angle of the cage may also be important factors affecting the sagittal alignment of the lumbar spine (Issa et al., 2023). ② The time of the post-operative MRI examination is different. We performed an MRI examination after the operation for at least one year, not immediately after surgery. As the follow-up time is prolonged, the ligamentum flavum thickness will decrease, the ligamentum flavum will remodel, and the spinal canal’s central canal area and anteroposterior diameter will increase (Mahatthanatrakul et al., 2020). ③ Roussouly classified the lumbar spine into four types based on an asymptomatic population. However, all patients in this study suffered from various lumbar degenerative disorders that could impact the measurements.

In our study, we found that the ΔDSA and ΔAPD in the upper and lower adjacent segments in the OLIF group were higher than those in the MIS-TLIF group, and multiple linear regression analysis showed that the ΔL3/4 DSA and ΔL5/S1 DSA were positively correlated with the ΔL4/5 ADH and ΔSLA. Singhatanadgige et al. (2022) believed that the ΔDSA and ΔAPD of the upper and lower adjacent segments increased significantly postoperatively. There was a correlation between ΔSLA after L4/5 single-level MIS-TLIF and ΔDSA and ΔAPD in the lower adjacent segment, and ΔDH was positively correlated with the change of ΔAPD. In the OLIF procedure, surgeons insert a large cage between the vertebrae to achieve indirect decompression by expanding the intervertebral space and shrinking the bulging intervertebral disc and ligament, and to gain a better recovery of DH and SLA. In addition to decompressing the corresponding surgical segments, the recovery of DH and SLA will also affect the spinal canal area of adjacent segments. This also provides some ideas for our follow-up research. By changing the size and position of the cage, which affects DH and SLA, the spinal canal area of the surgical segment and its adjacent segments can be further improved.

This study also has some limitations. Firstly, because this is a retrospective study, there will inevitably be some selection bias among the patients that comprise the two cohorts. Secondly, the number of cases is relatively small because the surgical segment is restricted to L4/5, and patients are strictly enrolled according to the criteria. Finally, although the radiological changes of ADD and AFD were detected, the follow-up period was also short, given the timing of the development of ASD, making it difficult for us to assess the progression of ASD. To better understand ASD, multi-center, large-data, and prospective studies with long-term follow-up are required.

Conclusions

For L4/5 single-level lumbar degenerative diseases, both OLIF and MIS-TLIF can improve the lumbar sagittal alignment and achieve good clinical results. Compared with MIS-TLIF, OLIF has advantages in improving lumbar sagittal balance and radiological parameters of adjacent segments, especially decreasing degeneration of intervertebral discs and facet joints in the early stages. Enhancing the height and angle of the surgical level changes the radiological parameters of adjacent segments. Long-term follow-up is still necessary to clarify whether there is any difference in the impact on ASD between the two surgeries.

Supplemental Information

Supplemental Information 1 MIS-TLIF raw data

Raw data in MIS-TLIF group including demographics of patients and clinical outcomes.

Supplemental Information 2 Pre-operative radiographic data in MIS-TLIF group

Pre-operative radiographic raw data (including ADH, PDH, FH, FA, SLA, LL, ADD and AFD ) in MIS-TLIF group.

Supplemental Information 3 Post-operative radiographic data in MIS-TLIF group

Post-operative radiographic raw data (including ADH, PDH, FH, FA, SLA, LL, ADD and AFD ) in MIS-TLIF group.

Supplemental Information 4 OLIF raw data

Raw data in OLIF group including demographics of patients and clinical outcomes.

Supplemental Information 5 Pre-operative radiographic data in OLIF group

Pre-operative radiographic raw data (including ADH, PDH, FH, FA, SLA, LL, ADD and AFD ) in OLIF group.

Supplemental Information 6 Post-operative radiographic data in OLIF group

Post-operative radiographic raw data (including ADH, PDH, FH, FA, SLA, LL, ADD and AFD ) in OLIF group.

Additional Information and Declarations

Competing Interests

Author Contributions

Human Ethics

Data Availability

The authors declare there are no competing interests.

Lantao Liu conceived and designed the experiments, prepared figures and/or tables, authored or reviewed drafts of the article, and approved the final draft.

Chicheng Ma performed the experiments, analyzed the data, prepared figures and/or tables, authored or reviewed drafts of the article, and approved the final draft.

Lianghai Jiang analyzed the data, authored or reviewed drafts of the article, and approved the final draft.

Longwei Chen analyzed the data, prepared figures and/or tables, and approved the final draft.

Xinpeng Zhou analyzed the data, prepared figures and/or tables, and approved the final draft.

Dechun Wang conceived and designed the experiments, authored or reviewed drafts of the article, and approved the final draft.

The following information was supplied relating to ethical approvals (i.e., approving body and any reference numbers):

This study was approved by the institutional review board of the ethics committee of Qingdao Municipal Hospital (Y044).

The following information was supplied regarding data availability:

Raw data is available in the Supplemental Files.

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
