# Peer review of "Early radiological adjacent segment changes following L4/5 fusion: a retrospective comparative study of oblique lateral interbody fusion and minimally invasive transforaminal lumbar interbody fusion"

_PeerJ, doi:10.7717/peerj.19918_

## Round 0.1 · original submission · Major Revisions

Reviewer 1 ·

Basic reporting

**Good Design**

- Literature references are acceptable.
- The tables and overall structure are well done.
- The English is clear and correct

Experimental design

ASD is a significant issue in the daily practice of spine surgery that is challenging to address. It is essential to compare various approaches to managing.
In this study, the authors aim to compare different methods used in the production (ASD). They analyze the radiological findings and the clinical follow-up; however, the timing of the post-operative evaluation and the average duration of the follow-up are not clearly defined.
Radiographics outcome were well design and a optimal way to compare clinical finding with radiographics finding but also to much measures could confuse the result and conclusion.

Validity of the findings

The study demonstrates a solid design; however, certain aspects require attention. The sample size is limited, consisting of only five cases of ASD, which may hinder the ability to draw firm conclusions. Additionally, enhancing the clarity of the radiological findings and improving the follow-up process would be beneficial. Addressing these areas could significantly strengthen the overall impact of the research.

Additional comments

At this moment, I have no additional comments to provide.

·

Basic reporting

The manuscript titled “Comparison of early radiological adjacent segment changes between oblique lateral interbody fusion and minimally invasive transforaminal lumbar interbody fusion” reported and compared the radiological changes of surgery segment and adjacent segment between OLIF and MIS-TLIF of Lumber4-5 in early-stage after surgery. It may provide a clinical observational basis for ASD, since ASD is always regarded as a long-stage clinical complication following lumbar spinal fusion surgery, subsequent outcome reports and analyses are expected.

Experimental design

no comment

Validity of the findings

no comment

Additional comments

Some minor issues:
1. lumber4-5 is recommended to be added to the title
2. Line 164-165 “The 2 cohorts were matched by sex, age, and segment.” How could you do this match? Can you give some details? And how to avoid select bias? And this is a retrospective study?
3. Line210-214 “The distance between the adjacent superior and inferior endplates was shortest in the planes of the anterior surfaces of the…” Here, what does shortest mean? Compared with what?
4. Line340-342 “little is known about the effects of OLIF and MIS-TLIF on adjacent segments, so ...” Here short-term changes should be strengthened, not simply say “the effects”
5. Line394 “We believed OLIF could slow the radiographic degeneration of adjacent segments” Here “slow” is not appropriate, maybe “not accelerate” or “better than MIS-TLIF” or something else…
6. I would like the authors could explain their principles for surgical methods selection. For L4-5 segment degeneration, while conservative treatment is ineffective, what is the basis for choose between OLIF and MIS-TLIF

---

## Round 0.2 · accepted · Accept

Thank you for addressing the comments from both reviewers in your revision, which I have reviewed. Your manuscript is now ready for publication.

·

Basic reporting

The revised manuscript reported and compared the radiological changes of surgery segment and adjacent segment between OLIF and MIS-TLIF of Lumber4-5 in early-stage after surgery. It may provide a clinical observational basis for ASD, since ASD is always regarded as a long-stage clinical complication following lumbar spinal fusion surgery; subsequent outcome reports and analyses are expected. This clinical study was reasonably designed, and the conclusion was reliable.

Experimental design

-

Validity of the findings

-

Additional comments

The items I have mentioned before have been completely revised and/or explained. Thank you for carefully revising and responding to the corresponding comments, which have made the manuscript which makes the manuscript more appropriate and readable.